# Leydig Cells in Immunocastrated Polish Landrace Pig Testis: Differentiation Status and Steroid Enzyme Expression Status

**DOI:** 10.3390/ijms23116120

**Published:** 2022-05-30

**Authors:** Piotr Pawlicki, Anna Galuszka, Laura Pardyak, Ryszard Tuz, Bartosz J. Płachno, Martyna Malopolska, Klaudia Dubniewicz, Ping Yang, Malgorzata Kotula-Balak, Kazimierz Tarasiuk

**Affiliations:** 1Center of Experimental and Innovative Medicine, University of Agriculture in Krakow, Redzina 1c, 30-248 Krakow, Poland; piotr.pawlicki@urk.edu.pl (P.P.); laura.pardyak@urk.edu.pl (L.P.); 2Department of Animal Anatomy and Preclinical Sciences, University Centre of Veterinary Medicine JU-UA, University of Agriculture in Krakow, Mickiewicza 24/28, 30-059 Krakow, Poland; anna.galuszka@urk.edu.pl; 3Department of Genetics, Animal Breeding and Ethology, Faculty of Animal Science, University of Agriculture in Krakow, Mickiewicza 24/28, 30-059, Krakow, Poland; ryszard.tuz@urk.edu.pl; 4Department of Plant Cytology and Embryology, Institute of Botany, Jagiellonian University in Krakow, Gronostajowa 9, 30-387 Krakow, Poland; bartosz.plachno@uj.edu.pl; 5Department of Pig Breeding, National Research Institute of Animal Production, Krakowska 1, 32-083 Balice, Poland; martyna.malopolska@iz.edu.pl; 6Department of Infectious Diseases of Animals and Food Hygiene, University Centre of Veterinary Medicine JU-UA, University of Agriculture in Krakow, Mickiewicza 24/28, 30-059 Krakow, Poland; klaudia.dubniewicz@urk.edu.pl (K.D.); kazimierz.tarasiuk@urk.edu.pl (K.T.); 7MOE Joint International Research Laboratory of Animal Health and Food Safety, College of Veterinary Medicine, Nanjing Agricultural University, Nanjing 210009, China; yangping@njau.edu.cn

**Keywords:** boar, GnRH immunocastration, Leydig cell, steroidogenic enzymes, relaxin-like proteins

## Abstract

Porker immunocastration against gonadoliberin (GnRH) secretion has been utilized since 2009; however, consumers are still skeptical of it. This is due to not having full information available on the problem of a boar taint, as well as a lack of research on morphological and molecular changes that may occur in the animal reproductive system and other body systems. The present study aimed to explore the functional status of steroidogenic Leydig cells of the testicular interstitial tissue in immunocastrated Polish Landrace pigs. Analyses were performed using Western blot, immunohistochemistry for relaxin (RLN), insulin-like 3 protein (INSL3), pelleted growth factor receptor α (PDGFRα), cytochrome P450scc, 3β- and 17β-hydroxysteroid dehydrogenases (3β-HSD, 17β-HSD), cytochrome P450arom, and 5α-reductase (5α-RED). Immunoassay ELISA was used to measure the androstenone, testosterone, and estradiol levels in the testis and serum of immunocastrates. We revealed disturbances in the distribution and expression of (i) RLN, indicating an inflammatory reaction in the interstitial tissue; (ii) INSL3 and PDGFRα, indicating alterations in the differentiation and function of fetal, perinatal, or adult Leydig cell populations; (iii) P450scc, 3β-HSD, 17β-HSD, P450arom, and 5α-RED, indicating disturbances in the sex steroid hormone production and disturbed functional status of Leydig cells; as well as (iv) decreased levels of androstenone, testosterone, and estradiol in testicular tissue and serum, indicating the dedicated action of Improvac to reduce boar taint at both the hypothalamic–hypophysis–gonadal axis and local level (Leydig cells). In summary, our study provides a significant portion of knowledge on the function of Leydig cells after immunocastration, which is also important for the diagnosis and therapy of testis dysfunction due to GnRH action failure and/or Leydig cell differentiational–functional alterations.

## 1. Introduction

Surgical castration (gonadectomy) in the early days of male piglet life is an animal husbandry procedure that has been practiced for centuries in agriculture worldwide. It is estimated that about 100 million piglets are castrated annually in the 25 countries of the European Union (EU) [1]. Castration prevents the formation of testicular androgen, 16-androstene steroids (primarily androstenone), which is a major component of boar taint and also reduces aggressive and sexually oriented mounting [2]. Castration has been increasingly viewed in relation to both economic considerations and animal welfare concerns. Because an unanesthetized surgical castration is a painful experience for the piglets, this surgery is announced to be banned in the EU; therefore, it is expected that the population of pigs that are not castrated is going to increase soon [3]. One alternative is an immunocastration that is performed by giving injections of a modified gonadoliberin (GnRH) component along with an adjuvant [4]. The GnRH vaccine (Improvac) induces an endogenous immune reaction that leads to a high level of GnRH antibodies approximately two weeks after the second vaccination [5]. The antibody titer against GnRH subsequently reduces lutropin (LH) secretion, thus resulting in decreased sex steroid synthesis in the interstitial Leydig cells of the testis. Furthermore, it was demonstrated that the GnRH vaccine affects the testicular development and functions of fattening boars [6]. Recently, Srisuwatanasagul et al. [7] reported that an adjustment to the time of first vaccination may lead to testis immaturity correlated with anti-Mullerian hormone expression (AMH), or it can affect the testicular length, histomorphometry, and expressions of estrogen synthase (P450arom) and AMH. Moreover, there are very diverse observations regarding the immunocastration effect on porker meat quality and fattening. Huber et al. [8] showed that immunization of animals effectively led to a reduction in boar-taint-causing compounds in adipose tissue. Furthermore, they observed that the immunization of the animals affected the physical and chemical composition of the organism, which was at the medium level compared to the entire male and surgically castrated. Their research also revealed that the timing of vaccination affects the quality of the meat. Therefore, the definition of an ideal immunization program could be used to meet the needs of a specific group of consumers. In another study, immunocastration led to a slight reduction of intramuscular fat but provided a similar or better fatty acid profile in terms of fat consistency [9]. Other authors found no differences in the meat quality after immunocastration and surgical castration [10]. It was demonstrated also that immunocastration caused large fat accumulation associated with an increase in the activity of lipogenic enzymes [11]. Immunocastration increased the growth rate and the intake of food compared to surgical castration [8]. Therefore, such boars required different nutrient regimens not fully implemented into breeding but resulting in a marked increase in the breeding cost. The above facts reflect that disadvantages, in addition to the advantages, of immunocastration still exist. On the other hand, gene modification has been proposed as a possible alternative to surgical castration [12]. The authors claim that perceived animal welfare may encourage public support for the gene editing of food animals. However, the potential risks of the technology (a major concern raised by over 80% of the study participants) need to be addressed and conveyed to the public in order to encourage perceived animal welfare in an equal way with human nutrition needs. The use of genetic markers in the selection of pigs that are free of boar taint offers a non-invasive, cost-effective, welfare-friendly, and potential long-term solution. Baes et al. [13] demonstrated that small adipose samples obtained by biopsy provide similar genetic parameters to those described in the literature for larger samples, and they are, therefore, a reliable performance test for boar taint in live breeding candidates. Of note, the amount of boar taint can vary dramatically between different breeds [10]. Therefore, further study is essential to complete this examination. Genes and quantitative trait loci related to boar taint have been identified. Some genes have their roles described, while others do not have a well-determined function yet [14]. Currently, analysis using gene-editing tools, such as CRISPR/Cas, is utilized. Of note, the gene-editing approach was successfully tested in sheep and goats (based on delayed puberty by the knockout of kisspeptin receptor 1, KISSR1 gene), while inactivation of the sex-determining region, SRY gene (involved in testis development), is under construction in pigs [15,16,17]. Thus, these strategies seem to be promising future solutions. Genetic selection with the use of high genetic correlations between androstenone and estradiol levels appears to be an effective criterion [18]. Of note, an increase in aggressive behavior was reported in such males with changes in endogenous hormone levels. 

Interestingly, in contrast to that of many other mammals, the boar testis possesses a large amount of interstitial tissue comprised mainly of Leydig cells. Its functionality can be assessed by secreting insulin-like 3 peptide (INSL3), which is crucial for testis descent [19]. It is well-established that two morphologically and functionally different Leydig cell populations exist in the testis of mammals. One develops prenatally (fetal Leydig cells, FLCs) and the second arises postnatally (adult Leydig cells, ALCs) [20]. Only humans and boars an additional wave of Leydig cell proliferation (perinatal), regressing shortly after birth, been described [21]. Leydig cell generations have different gene expression profiles, which indicate their origin from separate stem cells. The development of FLCs is chorionic-gonadotropin-independent, while ALCs are under LH control [22]. After birth, FLCs rapidly decrease in number; they are supposed to die or degenerate or be replaced by newly developed ALCs in the postnatal testis. However, morphometric analyses of the fetal and postnatal testis suggested that FLCs persist even in the adult testis [23]. Barsoum et al. [24] suggested that perturbation of FLC differentiation in the fetal period affects ALC differentiation later in puberty. 

Leydig cells are responsible for sex steroid hormone (androgens and estrogens) production. The initiation, maintenance, and reinitiation of spermatogenesis depend on androgen action, while estrogens are regulators of spermatogenesis (differentiation and survival of spermatogenic cells) [25]. Under physiological conditions, boar Leydig cells secrete a very high amount of estrogens. The estrogen concentration is even higher than those in sows in estrus [26]. Biologically inactive estrone sulfatase is the major estrogen occurring in the testicular vein as a result of steroid sulfatase and estrogen sulfotransferase activity. Currently, there is no explanation for this phenomenon. Unlike in other species, including humans, in the boar testis, P450 aromatase is expressed exclusively in Leydig cells; however, the estrogen synthesis in boar Leydig cells is not directly related to aromatization processes [27]. 

Control of steroid biosynthesis in Leydig cells is based on LH action. The Leydig cell expresses all the enzymes essential for the conversion of cholesterol to androgens and estrogens. The major pathways of sex steroid hormone synthesis are well-established, and the sequence of the responsible steroidogenic enzymes has been elucidated [28]. There are three main classes of steroidogenic enzymes: the cytochrome P450 heme-containing proteins (cytochrome P450 side-chain cleavage; P450scc, cytochrome P450 17α-hydroxylase C17-C20 lyase; P450c17 and cytochrome P450 aromatase; P450arom), the hydroxysteroid dehydrogenases (3β-hydroxysteroid dehydrogenase/Δ5-Δ4-isomerase; 3β-HSD and 17β-hydroxysteroid dehydrogenase; 17β-HSD), and reductase (5α-reductase; 5α-RED). The latter converts testosterone to more potent androgen dihydrotestosterone (DHT). DHT production has not been reported in boar testis [29]. On the other hand, a significant reduction in the epididymal 5α-RED gene expression in Large White × Polish Landrace boars with the blockage of androgen signaling in prenatal life has been revealed by us [30]. The amount of androstenone secreted by Leydig cells depends mainly on the activity of both 3β-HSD and 5α-RED. The modulation of androstenone production can be achieved by decreasing steroidogenic enzyme activity or increasing androstenone degradation. For the latter, studies on enriching feedstuffs in the last few weeks of fattening with bioactive components, e.g., fermentable carbohydrates, such as inulin from chicory root, which effectively reduce boar taint, mainly by acting on lower endogenous production of skatole, were undertaken [31]. Depending on the animal species, the biosynthesis of sex hormones proceeds along one or both Δ4 and Δ5 pathways, respectively. In the pigs, the Δ5 pathway is dominant [32]. Boar Leydig cells produce nine main steroids: four compounds classified as androgens (testosterone, epiandrosterone, androstanediol, and 19 nortestosterone), two precursors of androgens or estrogens (dehydroepiandrosterone and androstenedione), two estrogenic compounds (estrone and estradiol), and androstenone. 

Taking into account the unique and incompletely known differentiation, as well as the biochemical steroidogenic characteristics of boar Leydig cells, studies on immunocastrated boar testes were conducted. For this reason, markers of Leydig cell differentiation: INSL3, PDGFRα, and those of Leydig cell steroidogenic function: P450scc, 3β-HSD, 17β-HSD, P450arom, and 5α-RED were analyzed. In addition, the tissue inflammatory process was evaluated via RLN immunostaining and tissue secretion activity through the measurement of androstenone, testosterone, and estradiol levels. The results to be obtained might be useful in understanding how to molecularly control the Leydig cell function in light of active searching for substitutes for surgical castration, with a special emphasis on side effects at both the cellular and molecular levels.

## 2. Results

### 2.1. Topography, Morphology of the Testis, and RLN, INSL3, and PDGFRα Localization and Expression

Both topographic and morphological analyses revealed that immunocastrated testes are still spermatogenically active (Figure 1A,A’,B,B”). In some seminiferous tubules, open lumens with visible elongated spermatids were present, although in a small amount. Morphological analysis on semi-thin testicular sections showed shrinkage of the Leydig cells in immunocastrated boars (Figure 1A,B,B”).

Furthermore, the immunohistochemical localization of RLN showed positive immunosignal in the Leydig cells of control and immunocastrated boar testes (Figure 1C,D), with increased expression in the latter versus the controls (Figure 1G). Similarly, the presence of INSL3 was revealed in the Leydig cells of control and immunocastrated boar testes (Figure 1E,F), with decreased expression in immunocastrates (Figure 1G).

The positive signal for PDGFRα was revealed exclusively in Leydig cells of control and immunocastrate (Figure 2a(A,B)). Measurement of expression revealed decreased (*p* < 0.01 and *p* < 0.05, respectively) PDGFRα expression in immunocastrates (Figure 2b,c). 

### 2.2. mRNA and Protein Steroidogenic Enzyme Expression 

An analysis of the expression of P450scc, 3β-HSD, or 17β-HSD, P450arom, and 5α-RED at both the mRNA and protein levels revealed a significant reduction (*p* < 0.05, *p* < 0.01, *p* < 0.001) in the expression of transcripts and proteins in immunocastrates compared to controls (Figure 3a–e,a’–e’).

### 2.3. Steroidogenic Enzyme Expression and Localization 

In the control or immunocastrated testes, P450scc localization was observed in cells of the seminiferous tubules and interstitial tissue, with a higher amount of this enzyme in immunocastrated boars (*p* < 0.01) (Figure 4(aA,aA’,b)).

A very weak positive signal for 3β-HSD or 17β-HSD, P450arom, and 5α-RED was revealed in the immunocastrated testes (*p* < 0.01, *p* < 0.001, and *p* < 0.001) compared to controls, where the signals of moderate intensities were present in the Leydig cells (Figure 4(aB,aB’,aC,aC’,aD,aD’,aE,aE’,b)).

### 2.4. Testicular and Serum Androstenone, Testosterone, and Estradiol Levels 

In sera or testes of immunocastrates, significantly decreased levels of androstenone (*p* < 0.01, *p* < 0.001, respectively), testosterone (*p* < 0.01), and estradiol (*p* < 0.01) were revealed compared to the controls (Figure 5).

## 3. Discussion

Even though a vaccine for immunocastration has been available in the European Union since 2009, its practical use is limited due to a rather low market acceptance [33]. The opinion of consumers regarding immunocastration, who expect healthy, safe, and tasty meat, has not been thoroughly investigated, and they are mostly not well-informed regarding boar taint, the methods used to prevent it, as well as possible functional, molecular, and behavioral changes in porkers that resulted from vaccination against GnRH. An online questionnaire in 16 countries (>175 respondents/country) revealed that surgical castration without pain relief that was separated from each of the alternatives due to animal welfare showed the lowest acceptability (32%) [34]. Within the alternatives, a further partitioning between them was based on perceived quality and food safety, with an acceptance of 85% for applying anesthesia, 71% for immunocastration, and 49% for boar production. Differences depending on professional involvement and familiarity with agriculture were observed. The available detailed studies discovered important differences in the understanding and acceptance of immunocastration across Europe. For Swiss consumers, the most acceptable alternative was surgical castration with anesthesia/analgesia, while immunocastration was not favored [35]. Swedish, Belgian, and German consumers rather preferred meat from immunocastrates over the entire males and standard surgical castrates [36]. A survey with over 4000 consumers in France, Germany, and the Netherlands suggested that the fear of negative consumer attitudes towards immunocastration might be overestimated [37]. A recent study reported that Belgian farmers changed their opinion after having used different alternatives in practice and preferred entire males and immunocastration (anesthesia and/or analgesia were the least acceptable due to being the most demanding: labor-intensive, costly, and complex) [38]. According to the PIGCAS project survey, the scientists perceived immunocastration as a better alternative to surgical castration with anesthesia/analgesia due to being more practical and having benefits for animal welfare and economics [39]. Other drawbacks expressed by stakeholders were related to the ease of use in group-housing or outdoor production systems and security at work (fear of self-vaccination). Breeding entire boars kept in a healthy and socially stable environment with sufficient physical resources, as safeguarded by measures of enhanced animal care and management control without subsequent conventional surgical castration or immunocastration, seems to be a welfare-conforming, future-oriented action [3,40,41]. It is noteworthy that the feeding requirements of breeding whole male boars and behaviors of such boars need to be paid special attention. 

In this study, a slight increase in the fat thickness of immunocastrates was noted (M. Malopolska data and own observations; unpublished). Moreover, in our most recent studies, for the first time, the coincidence of disturbed adiponectin signaling and leptin signaling, together with increased cholesterol concentration and attenuated spermatogenesis, were reported [42]. Altered GnRH signaling affected the adipokine system in the testes of Landrace castrates. This probably induces further functional changes, leading to, e.g., complete spermatogenesis alteration, as well as lipid homeostasis and fattening perturbances.

Herein, we have shown that immunocastration results in changes in the cellular and molecular status of Leydig cells, affecting, possibly directly, the morpho-functional status of other tissues of the boar reproductive system or other organs, as well as boar behavior [6,11]. We additionally revealed inflammatory processes that take place in the testis through RLN delocalization and increased expression in immunocastrated boars. The involvement of RLN in inflammatory reactions is well-described in various organs and physiological conditions [42]. It plays an inhibitory role in the action of inflammatory cells and signals. Therefore, RLN seems to be one of several hormonal factors targeted by Improvac. Inflammatory mechanisms after Improvac treatment lead to severe seminiferous tubule epithelium disruption in Landrace/Yorkshire × Duroc/Pietrain boar testes [43] and Large White Land boar testes [6]. However, in the Polish Landrace boars studied here, only minor changes in spermatogenesis were found in the boars treated with Improvac according to the manufacturer’s instructions. Elongated spermatids and spermatozoa were still present in the seminiferous tubules. It seems possible that, due to various pig genetic backgrounds, adequate intensification of spermatogenesis may be observed. Furthermore, the effects of the genetic background on hormonal profiles may be reflected by the total inhibitory action also on GnRH in Landrace/Yorkshire × Duroc/Pietrain and Large White Land boar testes [6,44] but not in Polish Landrace boars. In the latter, the Improvac action seems to be partial. The injection of the vaccine is a systemic event that disturbs the hormonal homeostasis of the animal thus adverse effects could be expected in other tissues apart from the testes. A previous study has suggested that immunization against GnRH damages the hypothalamus [45]. In addition, findings from boars immunocastrated with GnRH antagonist, deslorelin, Polish Landrace × (Duroc × Pietrain) showed morphological and functional alterations in the testes and epididymis, although with a significant increase in spermatozoa production [46,47]. Saving animal reproductive function seems to be an important benefit of immunocastration that is revealed here in Landrace boars just after the full immunization procedure. In fact, GnRH immunocastration is reversible in time [5]. Therefore, it seems to be exciting to study further how genetic background regulates spermatogenesis recovery. Interestingly, in mares with endometriosis, repeated vaccinations with Improvac mitigated owner-reported behavioral abnormalities and stopped unilateral granulose theca cell tumor development [48]. Pregnancies have been reported in some management systems with slow-growing genotypes (e.g., under organic farming conditions) or heavy pig production [3]. In line with a morphological study by Stojanovic et al. [44], who reported shrinkage of the Leydig cell cytoplasm, nuclei, and the shrinkage of the interstitial tissue in immunocastrates, we found here an additional molecular alteration in the Leydig cells of these animals: expression status of differentiation markers and functional ones. It is possible that Improvac affects Leydig cell development and/or the exchange of FLC, perinatal, and ALC populations, respectively. It is confirmed by changes in the localization and expression of the Leydig cell differentiation markers INSL3 and PDGFRα. The hormone INSL3 is known to be expressed by terminally differentiated FLCs and ALCs [22]. The observed decreased INSL3 expression in the Leydig cells of immunocastrates may indicate that their interstitial tissue is dominated by FLC, which is probably due to a decrease in the production of sex hormones and a decreased level of LH. GnRH immunocastrates seem to be valuable material for future research on Leydig cell differentiation. Activation of the receptor by INSL3 induces differentiation of the gubernaculum, which pulls the testis caudally, thus driving the transabdominal phase of testicular descent [49]. Loss of *Insl3* causes cryptorchidism, or retention of the testes in the body cavity, which is associated with low serum testosterone and infertility, as well as a higher risk of developing testicular tumors [50]. 

PDFGFα signaling is a regulator of many developmental processes, including the formation of the male gonad [45]. PDGFRα is the only known marker of ALC stem cells that can self-renew and give rise to ALCs both in vivo and in vitro [51]. Found here, its increased expression may indicate the presence of ALC stem cells. Furthermore, *Pdgfra* knockout mice show that testes do not initiate the migration of cells from the mesonephros, fail to develop testis cords, and never develop FLCs in ex vivo organ cultures [52]. 

Morphological events occur before the determination of the sex. Steroidogenic factor-1 positive cells intermingling with primordial gonadal cells seem to proliferate and will become the gonad. The expression of steroidogenic factor-1 at the early stages of differentiation is required later for the expression of steroidogenic enzymes and constitutes the first sign of the FLC population [53]. The embryonic origin of FLCs is a topic of heated debate. Some studies suggest that the FLC population and the ALC population share a common progenitor pool. Perturbation of FLC development at the fetal stage induces ALC dysfunction in adults, suggesting a functional link between them. According to Griswold [21], some FLCs dedifferentiate at the fetal stage, serving as ALC stem cells. In contrast to ALCs, FLCs acquire the expression of steroidogenic enzymes concurrently or in very rapid succession [54]. The acquisition of steroidogenic enzymes in FLCs is a useful indicator to identify members of the FLC lineage, but the expression of individual enzymes cannot define different stages of FLC development. Increased or decreased expression or lack of expression of steroidogenic enzymes may be linked to FLC and/or perinatal population differentiation and functional disturbances, and/or their constant presence in the testes after birth and/or differentiation and functional disturbances in the ALC lineage in immunocastrated porcine testes. Kubale et al. [6] found a progressive decrease in the size and number of Leydig cells in Large White Land boars treated with Improvac. In addition, in deslorelin-treated boars, molecular disturbances in androgen receptor expression were demonstrated in the studied Leydig cell populations [55]. In the studies described here distinctly, molecularly opposite alterations were revealed in Leydig cells at the first step of cholesterol processing by P450scc and especially at the next steps of steroid hormone biosynthesis conducted by 3β-HSD, 17β-HSD, as well as P450arom and 5α-RED. The decreased mRNA, protein expression, or lack of protein presence are shown to correlate with levels of intratesticular and serum androstenone, testosterone, and estrogen in immunocastrates. In Large White Land boars treated with Improvac, a decreased immunoreactivity of 3β-HSD and LH receptors was found in correlation to a decreased level of androstenone and estradiol [6]. The authors also found that an increase in slaughter delay (2–8 weeks after second vaccination) did not result in signs of functional or morphological restoration of the immunocastrated testis. Improvac vaccination significantly improved the average daily gain compared to the non-treated group in male broiler chickens [56]. Concomitantly, the reproductive efficiency was significantly decreased, as well as the serum testosterone concentration, spermatogenesis, and expression levels of genes related to testosterone metabolism (*Cyp_17_ A_1_, Cyp_19_, HSD_3_ B_1_*, and *HSD_17_ B_3_*) and spermatogenesis (*Cyclin A_1_* and *Cyclin A_2_*) being significantly reduced in the immunized groups. In the present study, we found markedly reduced androstenone, testosterone, and estradiol concentrations. The skatole levels were also reduced by immunocastration [57,58]. Furthermore, androstenone and estradiol were identified as potential inhibitors of the expression and/or activity of the major skatole-metabolizing enzymes CYP2E1 and CYP2A [59]. The activities of skatole-metabolizing enzymes in the liver were higher in surgically or immunocastrated male pigs than in entire male pigs [60]. In the absence of androstenone and estradiol, the hepatic metabolism of skatole is not inhibited and produced skatole metabolites are readily eliminated from the body. Our results are in line with those by Dunshea et al. [57], who reported that, in immunocastrated male Large White x Landrace pigs, the fecal and serum testosterone concentrations decreased independent of vaccination scheme, e.g., in pigs slaughtered at 23 weeks of age and vaccinated at 15–19 weeks of age contrary to pigs slaughtered at 26 weeks age and vaccinated at 18–22 weeks of age. This also shows that GnRH vaccination is reversible. Additionally, vaccination with Improvac increased the growth rate and feed intake, although backfat also increased slightly. The incidence of combat injuries at slaughter was decreased markedly in pigs vaccinated against GnRH. These findings are supported by the results concerning behavioral observations of other authors [61]. The effect of immunocastration can last up to 22 weeks after the second injection [62]. On the contrary, Rottner and Claus [63] described individual differences in the resumption of testicular sex steroid production from 10 to 24 weeks after the second vaccination. Several studies have been conducted using alternative vaccination schemes. A study by Brunius et al. [64] demonstrated that, after early vaccination with Improvac applied to entire male pigs at 10 and 14 weeks of age (prepubertal or early pubertal), the levels of androstenone and skatole did not differ from the pigs vaccinated according to the manufacturer’s instructions. Furthermore, it was revealed that, already at 2 weeks following the second vaccination, the levels of androstenone and skatole were below the sensory threshold. 

A series of studies conducted at the beginning of the 21st century and subsequent research demonstrated that the proper androgen/estrogen balance is fundamental for normal male sexual development and function in humans and animals [65]. Variations in the balance of sex steroids or their activities are related to testicular disorders and infertility and are also a feature of the precocious aging paradox that is often observed in the last years. Findings obtained from animal studies might provide clues to the causative mechanisms of male reproductive dysfunctions, such as testicular dysgenesis syndrome, in humans, which is a global and growing problem. Therefore, it is of special importance to further understand the mechanisms, acting factors, and consequences that occur in the male reproductive system in reference to Leydig cell development and steroidogenic function. 

The present study also provides an important portion of the knowledge not only for boar breeders but also for basic scientists and clinicians on Leydig cell morpho-functional status after GnRH activity modulation. Expression changes in Leydig cell population differentiation markers, steroidogenic enzymes, and levels of produced sex steroid hormones are reported for the first time in Polish Landrace pigs and after GnRH immunization. These observations of an inflammatory reaction in the testicular interstitial tissue (RLN immunostaining analysis), alterations in differentiation and function of Leydig cell populations (INSL3 and PDGFRα analysis), changes in the functional status of Leydig cells (P450scc, 3β-HSD, 17β-HSD, P450arom, and 5α-RED analysis), and decreased levels of androstenone, testosterone, and estradiol in testicular tissue and serum, indicating the dedicated action of Improvac to reduce boar taint at both the hypothalamic–hypophysis–gonadal axis and Leydig cell level, are especially useful for porker producers and porker consumers. Moreover, these results may also be valuable in the diagnostics and treatment of Leydig-cell-dysfunction-related diseases in animals and humans.

## 4. Materials and Methods

### 4.1. Tissue Collection 

Testes were collected from n = 40 Polish Landrace boars at the Pig Slaughter Utility Control Station Chorzelow, Poland. Ten boars were kept untouched and used as control animals. Thirty boars were injected with Improvac (Zoetis, Louvain-la-Neuve, Belgium) according to the manufacturer’s recommendations (one dose (2 mL) after 8 weeks of age, the second one 4–6 weeks before slaughter at the age of 117 days). The animals at the Control Station were kept in individual pens, and the control fattening was performed from 30 to 120 kg (data in preparation). The feeding program was based on two mixtures I: proper fattening feed used from 30 kg to 80 kg (13.5 Mj/kg energy, 17–19% total protein, 2.4–4.0% crude fiber) and mixture II: feed at the end of fattening from 80 up to 120 kg (13 Mj/kg energy, 16–18% total protein, 3.0–5.0% fiber). Nutrition took place in the ad libitum system with a finely granulated mixture of a known and constant composition fed from vending machines. Feed intake was strictly controlled. The remaining conditions of keeping and handling the animals were in line with the current methodology of evaluation in Pig Slaughter Utility Control Station Chorzelow according to Directive 2010/63/EU on the Protection of Animals Used for Scientific Purposes. The use of boar tissues was approved by the Local Ethics Committee in Krakow, Poland (permission number: 144b/2015).

All animals vaccinated with Improvac showed a response. The skin, behavior, and breeding conditions were monitored twice a day. After animals (uncastrated boars and immunocastrated ones) were slaughtered, testicular tissues were obtained and cut into pieces and were fixed in 10% formalin or immediately frozen for laboratory analyses. Blood was collected from the jugular veins. 

### 4.2. Testis Topography and Morphology

Pieces of the testicular tissue were fixed in a mixture of 2.5% formaldehyde with 2.5% glutaraldehyde in a 0.05 M cacodylate buffer (Sigma-Aldrich; Saint Louis, MO, USA; pH 7.2) for several days, washed three times in a 0.1 M sodium cacodylate buffer, and later dehydrated and subjected to critical-point drying. Specimens were then sputter-coated with gold and examined at an accelerating voltage of 20 kV or 10 kV using a Hitachi S-4700 scanning electron microscope (Hitachi, Tokyo, Japan).

For morphology analysis, testicular tissues were fixed in 2.5% (*v*/*v*) glutaraldehyde 4% (*v*/*v*) formaldehyde in a 0.1 M sodium cacodylate buffer (pH 7.0) for several days, washed three times in a 0.1 M sodium cacodylate buffer pH 7, and post-fixed in a 1% (*w*/*v*) osmium tetroxide solution for 1.5 h at 0 °C. Dehydration using a graded ethanol series and infiltration and embedding using an epoxy embedding medium kit (Fluka, Buchs, Switzerland) followed. Semi-thin sections (0.9–1.0 μm thick) were prepared for light microscopy (LM) and stained for general histology using aqueous methylene blue/azure II (MB/AII) for 1–2 min and were examined using an Olympus B×60 light microscope.

### 4.3. Quantitative RT-PCR

Total RNA was extracted with TRIzol^®^ reagent (Life Technologies, Gaithersburg, MD, USA) according to the manufacturer’s instructions. Residual DNA was removed with TURBO DNA-free Kit (Ambion, Austin, TX, USA). The yield and quality of the RNA were evaluated by checking the A260:A280 ratio (NanoDrop ND2000 Spectrophotometer, Thermo Scientific, Rocheford, IL, USA) and by electrophoresis. High-Capacity cDNA Reverse Transcription Kit (Applied Biosystems, Carlsbad, CA, USA) was used to generate cDNA. For each RNA sample, reactions in the absence of RT were run to appraise genomic DNA contamination. RT-qPCR analyses were performed with the 10 ng cDNA templates, 0.5 μM primers (Institute of Biochemistry and Biophysics, Polish Academy of Sciences, (Appendix A), and SYBR Green master mix (Applied Biosystems, Carlsbad, CA, USA) in a final volume of 10 μL with the StepOne Real-time PCR system (Applied Biosystems, Carlsbad, CA, USA). PCR conditions: 55 °C for 2 min, 94 °C for 10 min, followed by denaturation temperature 95 °C for 15 s and annealing temperature for 60 s to determine the cycle threshold (Ct) for quantitative measurement. Amplification efficiency was between 97% and 104%. Melting curve analysis and agarose gel electrophoresis were used to confirm amplification specificity. Negative control reactions corresponding to RT reaction without the reverse transcriptase enzyme and a blank sample were carried out. The reference gene candidates were tested on experimental and control samples. The Microsoft-Excel-based application NormFinder was used to analyze the expression stability of commonly used reference genes. Based on these analyses, a housekeeping gene for normalizing RNA expression was selected. mRNA expressions were normalized to the mean expression of the reference genes (relative quantification, RQ = 1) with the use of the 2^−ΔΔCt^ method.

### 4.4. Western Blot

Testicular tissues were homogenized and sonicated with a cold Tris/EDTA buffer (50 mM Tris, 1 mM EDTA, pH 7.5) supplemented with inhibitors of serine, cysteine, acid proteases, and aminopeptidases (Sigma-Aldrich, Saint Louis, MO, USA). The protein concentration was estimated using the Bio-Rad DC Protein Assay Kit with BSA as a standard (Bio-Rad Labs, GmbH, München, Germany). Equal amounts of protein were resolved by SDS-PAGE under reducing conditions, transferred to polyvinylidene difluoride membranes (Merck Millipore, Darmstadt, Germany), and analyzed by Western blotting with antibodies (Appendix A). The presence of the primary antibody was revealed with horseradish peroxidase-conjugated secondary antibodies diluted at 1:3000 (Vector Laboratories, Burlingame, CA, USA) and visualized with an enhanced chemiluminescence detection system as previously described [20]. The specificity of antibodies was assessed using blocking peptide and/or antibody dilutant. All immunoblots were stripped with a stripping buffer containing 62.5-mM Tris-HCL, 100-mM 2-mercaptoethanol, and 2% SDS (wt: v; pH 6.7) at 50 °C for 30 min, and then incubated in antibody against β-actin (loading control). Protein abundance within the control group was arbitrarily set as 1, against which the statistical significance of the experimental groups was analyzed. To obtain quantitative results, the bands (representing each data point) were densitometrically scanned using the public domain ImageJ software (National Institutes of Health, Bethesda, MD, USA). The results of separate measurements were expressed as mean ± SD.

Separation of protein was performed by SDS-PAGE under reducing conditions and the transfer of proteins to polyvinylidene difluoride membranes. Nonspecific binding sites were blocked with non-fat dry milk containing Tween^®^ 20. Next, the membranes were incubated with antibodies (Appendix A) the same as for immunohistochemistry at 4 °C overnight, followed by a horseradish-peroxidase-conjugated secondary antibody (Vector Laboratories, Burlingame, CA, USA) at room temperature. Proteins were detected by chemiluminescence and documented with a ChemiDocTM XRS + System (Bio-Rad Labs. GmbH, München, Germany). The specificity of antibodies was assessed as positive controls (not shown). All immunoblots were stripped and re-probed with an anti-β-actin antibody as the loading control. The molecular weights of target proteins were estimated by reference to standard proteins (Sigma-Aldrich, Saint Louis, MO, USA). To obtain quantitative results, immunoblots were analyzed densitometrically using the ImageLab software (Bio-Rad Labs. GmbH, München, Germany) by an independent observer. 

### 4.5. Immunohistochemistry 

Testicular tissues were cut into 4 µm thin sections. Sections were immersed in 10 mM citrate buffer (pH 6.0) and heated in a microwave oven (2 × 5 min, 700 W). Thereafter, sections were incubated sequentially in H_2_O_2_ (3%; *v*/*v*) for 10 min and normal goat serum (5%; *v*/*v*) for 30 min as described previously [66]. Thereafter, the sections were incubated overnight at 4 °C with primary antibodies (Appendix A). Next, biotinylated goat anti-rabbit IgG (Vector, Burlingame, CA, USA) and avidin-biotinylated horseradish peroxidase complex (ABC/HRP; Dako, Glostrup, Denmark) were applied in the succession. The staining was developed using 3,3′-diaminobenzidine (DAB). Positive controls (mouse primary Leydig cells; for isolation procedure, please see Ref. [67]) and negative controls (in the absence of primary antibodies) were performed for each immunostaining. Thereafter, sections were slightly counterstained with Mayer’s hematoxylin and mounted using DPX mounting media (Sigma-Aldrich, Saint Louis, MO, USA). Serial sections were examined with a Leica DMR microscope (Leica Microsystems, Wetzlar, Germany). 

With the use of ImageJ software (National Institutes of Health, Bethesda, MD, USA), minimum 50 cells per section were analyzed by outlining a profile of the individual immunoreactive regions. Densitometric reading (optical density; OD) taken from all cells in each section was then combined and averaged to obtain the total OD for that section. The background OD staining of each section was measured by averaging four random immunonegative areas over the image of the optic chiasm. The true OD for each section was then expressed by subtracting the background OD from the total OD so that each measurement was made in an unbiased way to correct for the background. For calculations of the intensity of immunohistochemical reaction for an individual protein expressed as relative optical density (ROD) of diaminobenzidine brown reaction products, the following formula was used:ROD = OD_specimen_/OD_background_ = log GL_blank_/GL_specimen_/log GL_blank_
where the gray level (GL), respectively, is for specimen (stained area; GL specimen), for background (unstained area; GL background), and for blank (GL blank measured after the slide was removed from the light path that is equal to GL background; log GL_blank_ = GL_background_).

### 4.6. Androstenone, Testosterone, and Estradiol Concentration Measurement

For androstenone, testosterone, and estradiol concentration measurement, ELISA Assay (cat. No. ELK8586; cat. No. ELK1332 and cat. No. K1208; ELK-Biotechnology, Wuhan, China) were used according to the manufacturer’s instructions. Testicular lysates were obtained by homogenization and sonication of fresh tissues with a cold Tris/EDTA buffer (50 mM Tris, 1 mM EDTA, pH 7.5). Before assaying, sample amount was calculated based on the predicted concentration of hormone in testicular tissue or serum. For analysis, 50 µL standard (prepared according to instruction) or sample (testicular lysate or serum) were used. The blank was set with standard diluent buffer. Concentrations were calculated from mean ± SD (from three separate measurements). 

### 4.7. Statistics

Each variable was tested using the Shapiro–Wilk W-test for normality. The homogeneity of variance was assessed with Levene’s test. The distribution of the variables was normal and the values were homogeneous in variance. All statistical analyses were performed using a one-way analysis of variance (ANOVA) followed by Tukey’s post hoc comparison test to determine which values differed significantly from the controls. The analysis was conducted using Statistica software (StatSoft, Tulsa, OK, USA). Data were presented as a mean ± SD. Data were considered statistically significant at *p* < 0.05. All experimental measurements were performed in triplicate from material derived from different animals.

## Figures and Tables

**Figure 1 ijms-23-06120-f001:**
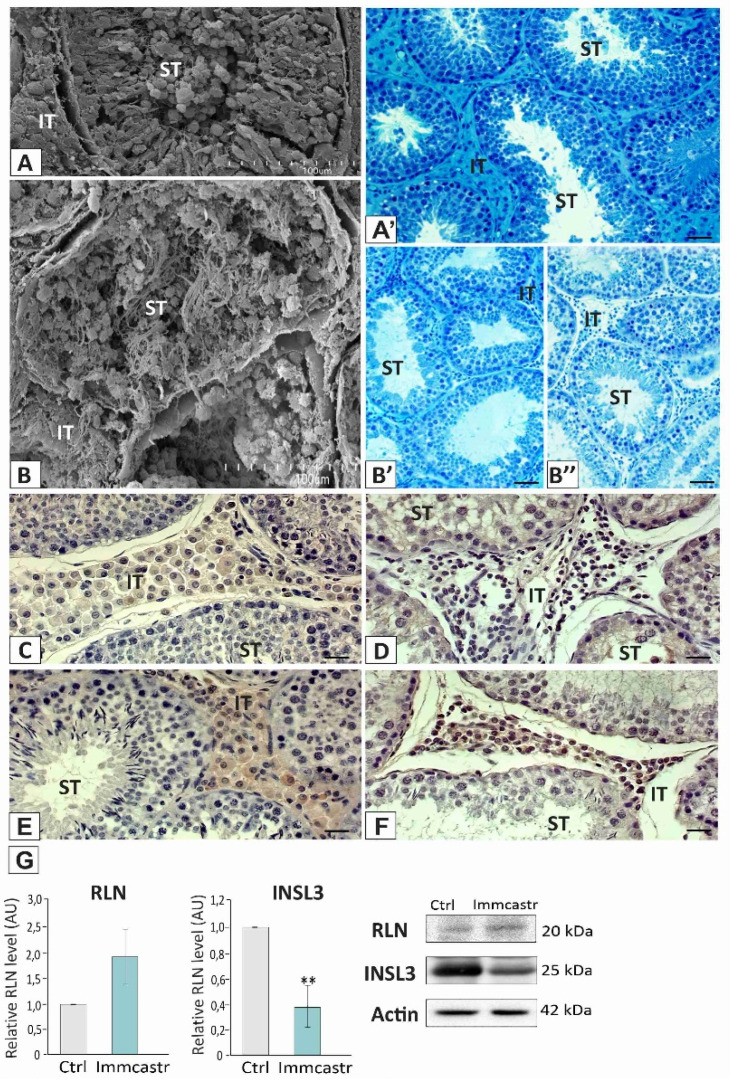
**Upper panel:** topography (**A**,**B**) and morphology (**A’**,**B’**,**B”**) of control and immunocastrated testes. Scanning electron microscopy, bar 45 µm; and light microscopy (methylene blue/azure II staining), bar 20 µm. ST-seminiferous tubule, IT-interstitial tissue. Analysis/staining was performed in three serial sections from each animal. **Lower panel:** representative microphotographs of immunohistochemical localization of RLN and INSL3 in control (**C**,**D**) and immunocastrated (**E**,**F**) testes. Bar 20 µm. Staining with DAB and counterstaining with hematoxylin. ST-seminiferous tubule, IT-interstitial tissue. Staining was performed in three serial sections from each animal. (**G**) Histograms with the quantitative representation of data (mean ± SD) of three independent experiments, each in triplicate, and Western blot detection of RLN and INSL3 proteins. The relative level of the studied protein was normalized to β-actin. The protein levels within the control group were arbitrarily set at 1. The histograms are the quantitative representation of data (mean ± SD) of three independent experiments, each in triplicate. Significant differences in relative optical density (ROD) of individual protein from control values are denoted as ** *p* < 0.01.

**Figure 2 ijms-23-06120-f002:**
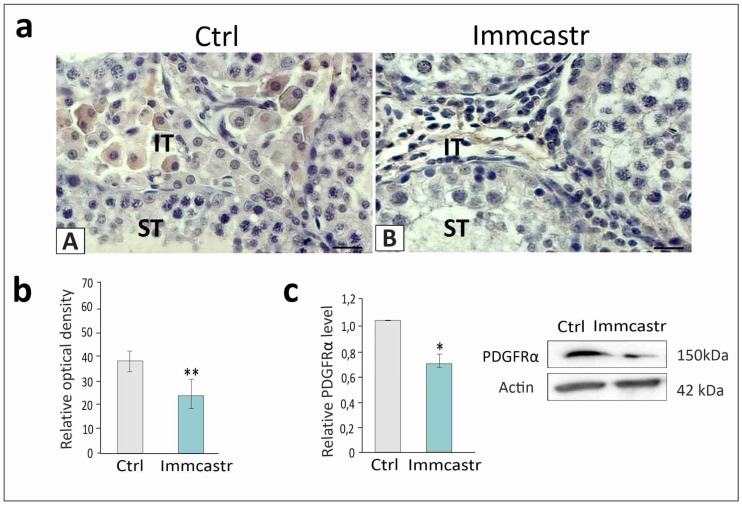
**Upper panel:** (**a**) representative microphotographs of immunohistochemical localization of PDGFRα in control (**A**) and immunocastrated (**B**) testes. Bar 20 µm. Staining with DAB and counterstaining with hematoxylin. ST-seminiferous tubule, IT-interstitial tissue. Staining was performed in three serial sections from each animal. **Lower panel**: (**b**) histograms with the quantitative representation of data (mean ± SD) of three independent experiments, each in triplicate, and (**c**) histograms with the quantitative representation of data (mean ± SD) of three independent experiments, each in triplicate, and Western blot detection of PDGFRα protein. The relative level of the studied protein was normalized to β-actin. The protein levels within the control group were arbitrarily set at 1. The histograms are the quantitative representation of data (mean ± SD) of three independent experiments, each in triplicate. Significant differences in relative optical density (ROD) of individual protein from control values are denoted as * *p* < 0.05 and ** *p* < 0.01.

**Figure 3 ijms-23-06120-f003:**
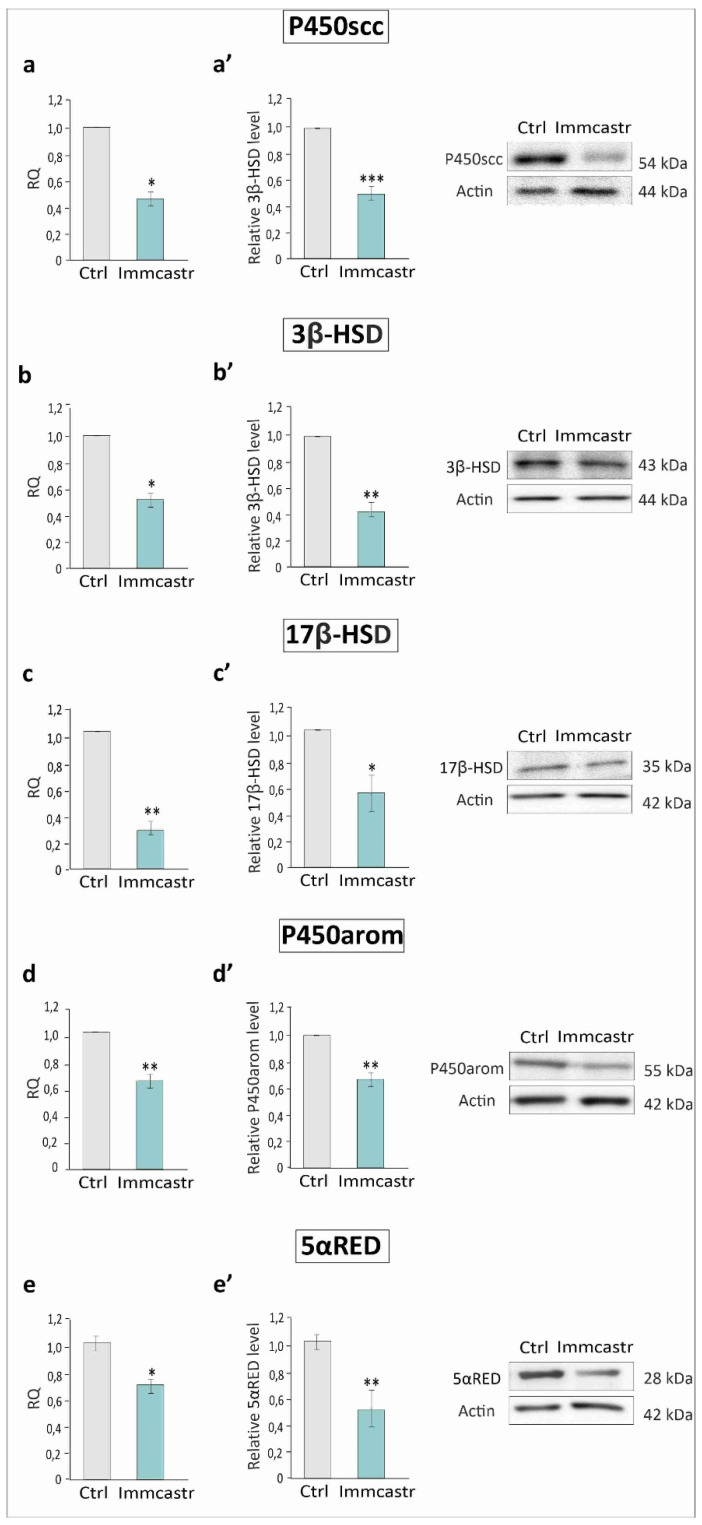
**Left panel:** (**a**–**e**) Relative expression of mRNAs (RQ) for P450scc, 3β-HSD, 17β-HSD, P450arom, and 5α-RED. The expression values of the individual genes were normalized to the mean expression of the reference gene. **Right panel:** (**a’**–**e’**) histograms with the quantitative representation of data (mean ± SD) of three independent experiments, each in triplicate, and Western blot detection of P450scc, 3β-HSD, 17β-HSD, P450arom, and 5α-RED proteins (three independent experiments, each in triplicate). The relative level of the studied protein was normalized to β-actin. The protein levels within the control group were arbitrarily set at 1. Significant differences from control values are denoted as * *p* < 0.05, ** *p* < 0.01, and *** *p* < 0.001.

**Figure 4 ijms-23-06120-f004:**
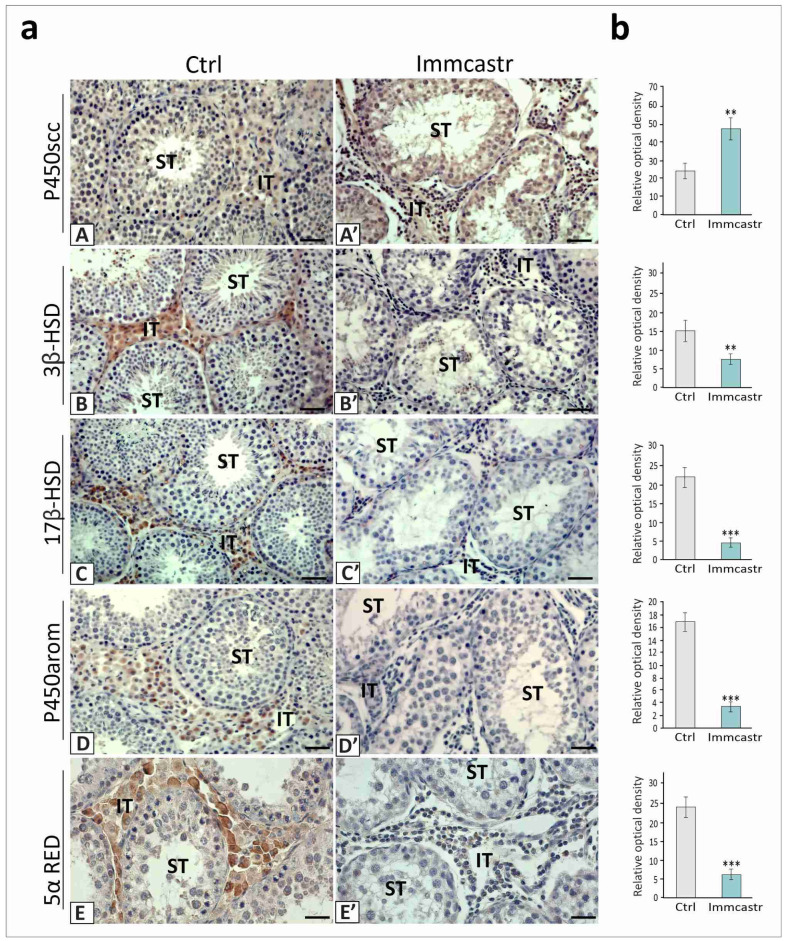
**Left panel:** (**a**) representative microphotographs of immunohistochemical localization of P450scc, 3β-HSD, 17β-HSD, P450arom, and 5α-RED in the control (**A**–**E**) and immunocastrated (**A’**–**E’**) testes. Bar 20 µm. Staining with DAB and counterstaining with hematoxylin. ST-seminiferous tubule, IT-interstitial tissue. Staining was performed in three serial sections of each animal. **Right panel:** (**b**) the histograms are the quantitative representation of data (mean ± SD) of three independent experiments, each in triplicate. Significant differences in relative optical density (ROD) of individual protein from control values are denoted as ** *p* < 0.01 and *** *p* < 0.001.

**Figure 5 ijms-23-06120-f005:**
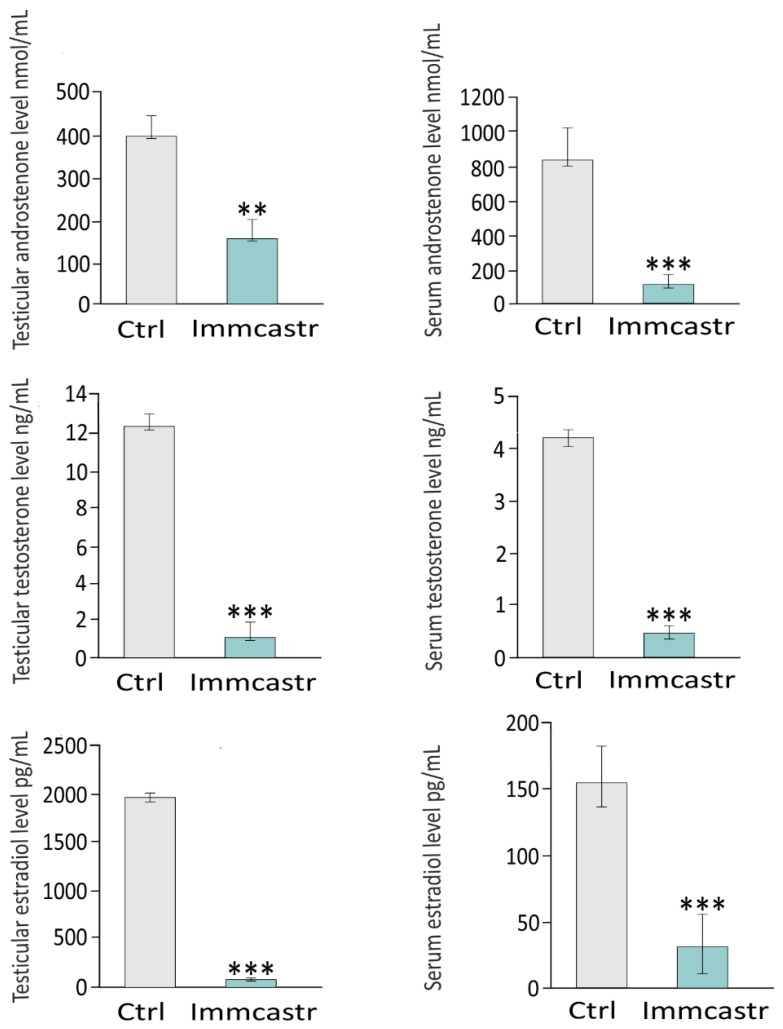
**Left panel:** androstenone, testosterone, and estradiol testicular level. **Right panel:** androstenone, testosterone, and estradiol serum level in control and immunocastrated testes. Data are expressed as means ± SD. Analyses were performed in triplicate. Significant differences from control values are denoted as ** *p* < 0.01 and *** *p* < 0.001.

## Data Availability

Data are contained within the article or Appendix A.

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
