# Peer review of "Leydig Cells in Immunocastrated Polish Landrace Pig Testis: Differentiation Status and Steroid Enzyme Expression Status"

_ijms, 2022, doi:10.3390/ijms23116120_

Round 1

Reviewer 1 Report

The manuscript titled “Leydig cells in…..status” describes an elegant study developed on a very up-to date topic: swine immunocastration in order to avoid surgical castration which is considered now an animal welfare concern.

The study is very interesting, well designed, clearly described and the manuscript is well written.

Introduction and discussion are particularly good: the first going deep in the issue without be boring and the latter commenting extensively the interesting results obtained.

I was very impressed by the study and by the variety of appropriate methods employed on a satisfactory number of samples. However, before be suitable for publication some revisions are needed: a couple important and one minor (see below).

    • Line 80: delete the comma after “Of”
    • RESULTS and MAT-METH: in the results Lines 139-148 and again further on in the text authors write about “increased expression”, “decreased expression” but a clear chapter dedicated to the assessment of immunohistochemical results is not reported in the manuscript. Something about is written in the images legends but is not enough. Authors have to state which method of assessment was used, which score/scale, quantitative, semi-quantitative? Stain intensity?
    • Ok for negative controls. Authors also consider positive controls. However, which were the tissue structures used as positive control of the immunohistochemical reactions? Please list them antibody by antibody
    •  

Author Response

Answers to questions by Reviewer #1 highlighted in red in the manuscript text 

1.    Line 80: delete the comma after “Of”
It was correct. Please see line:111.

2.    RESULTS and MAT-METH: in the results Lines 139-148 and again further on in the text authors write about “increased expression”, “decreased expression” but a clear chapter dedicated to the assessment of immunohistochemical results is not reported in the manuscript. Something about is written in the images legends but is not enough. Authors have to state which method of assessment was used, which score/scale, quantitative, semi-quantitative? Stain intensity?
With the use of ImageJ software (National Institutes of Health, Bethesda, MD, USA)
minimum 50 cells per section were measured by outlining a profile of the individual immunoreactive regions. Densitometric reading (optical density; OD) taken from all cells in each section was then combined and averaged to obtain the total OD for that section. The background OD staining of each section was measured by averaging four random immunonegative areas over the image of the optic chiasm. The true OD for each section was then expressed by subtracting the background OD from total OD, so that each measurement was made in an unbiased way to correct for the background. For calculations of the intensity of  immunohistochemical reaction for individual protein expressed as relative optical density (ROD) of diaminobenzidine
brown reaction products, the following formula was used:

ROD = ODspecimen/ODbackground = log GLblank/GLspecimen/log GLblank

where the gray level (GL) respectively is for specimen (stained area; GL specimen), for background (unstained area; GL background), and for blank (GL blank measured
after the slide was removed from the light path that is equal to GL background; log GLblank = GLbackground).

Additional details were included in the immunohistochemistry description. Please see lines:197-198, 213, 242-243, 538-555 and 800-803.

3.    Ok for negative controls. Authors also consider positive controls. However, which were the tissue structures used as positive control of the immunohistochemical reactions? Please list them antibody by antibody
As a positive control mouse, Leydig cells were used and stained for all antibodies. This information was included in the manuscript text. Please see lines:  532-533.

Reviewer 2 Report

In the present study, the effects of immunocastration against GnRH on Leydig cell morphology and function and steroid enzyme expression were investigated in Polish White Lop-Eared boars vaccinated with Improvac. The production of immunocastrates as an emerging alternative to surgical castration of male piglets has led to a number of new challenges, primarily related to detection and reduction of boar taint, carcass and meat quality, nutritional and housing conditions, reduction of aggression, animal welfare and consumer-related issues. While these tasks were extensively researched by animal scientists in recent years, the molecular aspects of immunocastration are less known and in this sense the present study is welcome. In addition, the study benefits from very advanced analytical methods and approaches.

However, from an animal science perspective, there are some parts of the study that are less clear and require further attention.

Rows 66 to 70 – This paragraph does not seem correct to me, as immunocastrates tend to have a higher meat content compared to surgical castrates and have an advantage over castrated males in terms of feeding efficiency.  Please, reconsider these statements.  

Rows 71 – 81 – This whole chapter seems to be written superficially because several leading international pig breeding companies have already included the problem of boar taint in their breeding programs and have successfully reduced this problem with their terminal boar lines, whose semen is already commercially available.

Row 85 and 525 – Please provide the full reference.

Rows 123 - 124 – The addition of fermentable carbohydrates, such as inulin from chicory root, to the diet in the last weeks of fattening can effectively reduce boar taint, mainly by acting on lower endogenous production of skatole. Please reconsider.

Rows 218 – 222 – Recently, several consumer- and stakeholder-oriented international studies on the acceptability of alternatives to surgical castration, including immunocastration, have been published in the journals of WOS, and the results of these studies should also be considered in the discussion.

Row 241 – Were there animals that did not show an immune response? Did you perform any post-vaccination monitoring?

Rows 245 – 247 and 254-255 - Please align the font size.

Rows 254-255 – It is known that the effect of immunization against GnRH decreases with time and the possibility of recovery of testicular function increases, i.e. immunocastration is reversible.

Row 310 – You refer to male broiler chicken here, is this correct? The reference paper title on page 584 should be provided.  

Rows 327-329 – Growth performance and backfat thickness and incidence of injury at slaughter are mentioned here, but no data on results or methodology are given in the text. How come?

Row 352-375 – Which animals were the control group in the study? How old were the immunocastrated animals at the time of the second vaccination?

Row 476 – Ad space (in variance)

Author Response

Answers to questions by Reviewer #2 highlighted in blue in the manuscript text 

1.    Rows 66 to 70 – This paragraph does not seem correct to me, as immunocastrates tend to have a higher meat content compared to surgical castrates and have an advantage over castrated males in terms of feeding efficiency.  Please, reconsider these statements.  
According to the suggestion, more information based on literature data was provided. Please see lines:77-78, 81-85, and 630-638.

2.Rows 71 – 81 – This whole chapter seems to be written superficially because several leading international pig breeding companies have already included the problem of boar taint in their breeding programs and have successfully reduced this problem with their terminal boar lines, whose semen is already commercially available.
According to the Reviewer’s suggestion, extensive information on boar taint studies was included. Please see lines:95-110 and 641-642, and 645-652.

3.Row 85 and 525 – Please provide the full reference.
It was provided. Please see line:657.

4. Rows 123 - 124 – The addition of fermentable carbohydrates, such as inulin from chicory root, to the diet in the last weeks of fattening can effectively reduce boar taint, mainly by acting on lower endogenous production of skatole. Please reconsider.
Reviewer suggestion was included in ms text. Please see lines:154-156.

5.Rows 218 – 222 – Recently, several consumer- and stakeholder-oriented international studies on the acceptability of alternatives to surgical castration, including immunocastration, have been published in the journals of WOS, and the results of these studies should also be considered in the discussion.
It was provided as suggested. Please see lines:266-287 and 706-722.

6.Row 241 – Were there animals that did not show an immune response? Did you perform any post-vaccination monitoring?
All animals vaccinated with Improvac showed a response. The skin, behavior, and breeding conditions were monitored twice a day. During the time of Improvac vaccination growth performance and backfat thickness of porkers were observed (data in preparation non-published). Information was included in the manuscript text. Please see lines:445-447.
7.Rows 245 – 247 and 254-255 - Please align the font size.
It was a result of pdf preparation. Word doc file will be sent together with a pdf file.

8.Rows 254-255 – It is known that the effect of immunization against GnRH decreases with time and the possibility of recovery of testicular function increases, i.e. immunocastration is reversible.
Clear information was provided as suggested. Please see lines:325-329.

9.Row 310 – You refer to male broiler chicken here, is this correct? The reference paper title on page 584 should be provided.  
It was corrected. Please see line:382 and 770-772, 800-803.

10.Rows 327-329 – Growth performance and backfat thickness and incidence of injury at slaughter are mentioned here, but no data on results or methodology are given in the text. How come?
During the time of Improvac vaccination growth performance and backfat thickness of porkers were observed (data in preparation non-published). Information was included in the manuscript text. Please see lines: 435, 445-447.

In addition, our last studies (under review) where we have shown for the first time the coincidence of disturbed adiponectin signaling and leptin signaling together with increased cholesterol concentration and attenuated spermatogenesis as a result of halted androstenone production. Altered Disturbed GnRH signaling action affects the adipokine system in testes of Landrace pigs castrates which may impact further functional changes leading to e.g. complete spermatogenesis alteration, as well as lipid homeostasis and fattening perturbances. Kotula-Balak M et al., Vaccination against gonadoliberin with Improvac influences adiponectin and leptin regulation in testes of Landrace pigs. Vet Med Science & Practice (under review)  
Information was included in the manuscript text. Please see lines: 293-299, and 728-731.

11.Row 352-375 – Which animals were the control group in the study? How old were the immunocastrated animals at the time of the second vaccination?
Information about control animals was corrected. Please see lines:430-431.
Animals were  117 days old time of the second vaccination. It was added. Please see line:433.

12. Row 476 – Ad space (in variance)
It was added. Please see line:572.

Round 2

Reviewer 2 Report

I thank the author for the changes and clarifications. However, although the revised manuscript has been significantly improved, it is still inaccurate in some parts. For example, the authors' interpretation of the Huber et al. (2018) results on the differences in carcass yield and meat percentage between immunocastrated and surgically castrated animals appear to be incorrect because in the original study, dressing percentage was greater in early castrated males than in late castrated and entire males (P < 0.05), whereas intermediate values were observed in immunocastrates. In addition, there were no differences in the proportion of dissected muscle to cold side weight between treatments. Thus, interpretation and discussion of issues related to animal performances and carcass and meat quality require more attention and understanding.  

Author Response

1. The authors' interpretation of the Huber et al. (2018) results on the differences in carcass yield and meat percentage between immunocastrated and surgically castrated animals appear to be incorrect because in the original study, dressing percentage was greater in early castrated males than in late castrated and entire males (P < 0.05), whereas intermediate values were observed in immunocastrates. In addition, there were no differences in the proportion of dissected muscle to cold side weight between treatments. Thus, interpretation and discussion of issues related to animal performances and carcass and meat quality require more attention and understanding. 

According to the Reviewer's suggestion extensive information on results reported in studies by Hubner et al., was added.